# Accelerated body size evolution in upland environments is correlated with recent speciation in South American freshwater fishes

Felipe O. Cerezer [1,2] ✉, Cristian S. Dambros[2], Marco T. P. Coelho [1], Fernanda A. S. Cassemiro [3], Elisa Barreto[1], James S. Albert[4], Rafael O. Wüest[1] & Catherine H. Graham[1]

Speciation rates vary greatly among taxa and regions and are shaped by both biotic and abiotic factors. However, the relative importance and interactions of these factors are not well understood. Here we investigate the potential drivers of speciation rates in South American freshwater fishes, the most diverse continental vertebrate fauna, by examining the roles of multiple biotic and abiotic factors. We integrate a dataset on species geographic distribution, phylogenetic, morphological, climatic, and habitat data. We find that Late Neogene-Quaternary speciation events are strongly associated with body-size evolution, particularly in lineages with small body sizes that inhabit higher elevations near the continental periphery. Conversely, the effects of temperature, area, and diversity-dependence, often thought to facilitate speciation, are negligible. By evaluating multiple factors simultaneously, we demonstrate that habitat characteristics associated with elevation, as well as body size evolution, correlate with rapid speciation in South American freshwater fishes. Our study emphasizes the importance of integrative approaches that consider the interplay of biotic and abiotic factors in generating macroecological patterns of species diversity.

Speciation, the process of lineage splitting, varies remarkably across the Tree of Life and contributes to uneven species diversity on Earth. Some groups, such as cichlid fishes in Lake Victoria, Africa, radiated into more than 500 species in only 15,000 years, whereas coelacanths in the Indo-West Pacific have produced only two species over approximately 80 million years ago (Mya)[1,2]. Scientists have long been fascinated by the differences in speciation rates across taxa and among regions[3], yet the main drivers of these differences remain hotly debated[4].

From a macroevolutionary perspective, speciation arises from the complex interplay of biotic and abiotic factors[5,6]. Biotic factors include mechanisms associated with morphological evolution and feedback loops resulting in diversity-dependent speciation[7–9]. Morphology is closely related to the use of resources and habitats[10]. When morphological traits evolve rapidly (i.e., high rates of trait change), speciation rates are expected to increase because of the exploration and partitioning of a broad spectrum of available resources[11–13]. Several studies have reported pulses of morphological evolution, followed by

[1]Swiss Federal Research Institute for Forest, Snow, and Landscape (WSL), Birmensdorf, Switzerland. [2]Programa de Pós-Graduação em Biodiversidade Animal, Departamento de Ecologia e Evolução, Universidade Federal de Santa Maria, Santa Maria, Brazil. [3]Programa de Pós-Graduação em Ecologia e Evolução, Universidade Federal de Goiás, Goiânia, Brazil. [4]Department of Biology, University of Louisiana at Lafayette, Lafayette, LA, USA. ✉e-mail: cerezerfelipe@gmail.com

accelerated speciation rates[7,8,14]. In contrast, other studies demonstrate that morphological differentiation can also be decoupled from speciation[15,16]. In addition to rates of morphological evolution, diversity-dependent mechanisms posit that the number of species locally co-occurring influences the dynamics of speciation rates[9]. Species diversity can slow speciation rates due to interspecific competition for resources[17] or increase speciation rates due to the greater potential for species interaction[18].

Abiotic factors, such as climate and physical habitat characteristics, can also affect speciation rates[5,6]. Variations in climatic conditions may drive speciation through niche divergence[19] (e.g., ecological speciation) and niche conservatism[20] (e.g., the barrier of unsuitable climate). Similarly, abiotic characteristics such as habitat volume and elevation can positively influence speciation rates because larger habitats and habitats at higher elevations are likely to be isolated by geographic barriers, thus creating opportunities for allopatric speciation[21]. In addition, regions with structural habitat heterogeneity (e.g., with more habitat or other resource patchiness) can promote ecological opportunities and trigger divergent adaptations that enhance speciation[3]. These biotic and abiotic factors are not mutually exclusive—species morphology is likely to evolve as a response to climate and habitat, which might, in turn, promote speciation[6]—but the interplay between these factors has been poorly studied.

Studies evaluating the relative contributions of biotic and abiotic mechanisms to speciation rates have frequently focused on single predictor variables and overlooked geographic dimensions[13,22]. Here we evaluate the association of multiple biotic and abiotic mechanisms with the speciation rates of the megadiverse fish fauna of South America (Table 1), which comprises at least 5160 species with remarkable ecological, morphological, and behavioral diversity[23]. Previous research investigating the factors driving speciation in freshwater fish has primarily relied on lineage-scale studies. More broadly, historical and geological events coincide with major shifts in lineage diversification of freshwater fishes[24–26]. In addition, high rates in the evolution of traits, such as body size[7,8] or reproductive mode[27], are often associated with rapid species accumulation. However, the spatial dynamics of speciation rates in this fauna are poorly understood, particularly the extent to which species traits and abiotic factors interact to influence speciation in different regions and clades.

We aim to understand the extent to which biotic and abiotic factors influence speciation rates across geographic regions. Our study concentrates on recent speciation events primarily taking place during the Late Neogene-Quaternary (mean species age of 7.3 Mya). This period holds paramount importance because it coincides with a better understanding of the paleoclimates and paleoenvironments of northern South America[28, 29], while also experiencing generally low extinction rates for most taxa of South American freshwater fishes[23]. We analyze speciation rates and eleven predictors in 460 drainage sub-basins, encompassing 2638 fish species. Here, we show how the evolution of several morphological traits, species diversity (diversity-dependence), climate, and characteristics of the physical habitat interact with spatial variation in speciation rates.

## Results

### Freshwater fish speciation across phylogeny and geography

Fish speciation in South America steadily increased over time, with a particularly notable uptick starting in the late Cretaceous period (Supplementary Fig. 1). Specifically, speciation rates varied over 18-fold across the tips of the phylogeny, with the highest rates being concentrated in certain groups such as the cyprinodontiform *Orestias*, the characiform *Serrasalmus*, and the siluriforms *Hemiancistrus*, *Pterygoplichthys*, and *Hypostomus* (Fig. 1a).

When mapped onto geography, recent speciation rates varied >17-fold among sub-basins (Fig. 1b). Lineages with the highest recent speciation rates were found in regions of the Altiplano and Pacific slope of Peru, mesic areas in the Atlantic Forest, and some seasonally dry upland regions (e.g., dry diagonal) (Fig. 1b). In contrast, the Southern Cone, Northeast Atlantic slope of Brazil, and some parts of the highly diverse Central and Eastern Amazon lowlands had the lowest speciation rates (Fig. 1b). We did not find any clear latitudinal (Pearson's $r = 0.161$, $p < 0.001$) or longitudinal trends ($r = 0.108$, $p = 0.020$) in speciation rates across South America.

### Correlates of spatial variation in speciation rates

Multiple regression analysis, including both biotic and abiotic variables, such as rates of morphological evolution, species diversity, climate, and habitat, explained 90% of the spatial variation in speciation rates (Fig. 1b, d; Fig. 2a; Supplementary Table 1).

Rates of morphological evolution accounted for 69% of the total variance in speciation rates across sub-basins (Fig. 2b). Among the various morphological traits examined, the rate of maximum body length evolution emerged as the most influential factor explaining

**Table 1 | Predicted drivers of speciation rates**

| General hypothesis | Mechanism of speciation | Variables | Predictions | References |
|---|---|---|---|---|
| Biotic | Rates of morphological evolution | Rates of body elongation evolution | High rates of morphological evolution of traits that are relevant for fish feeding ecology, physiology, and behavior can trigger speciation rates by enabling the exploration and partitioning of available resources | 7, 11, 37 |
| | | Rates of maximum body length evolution | | |
| | | Rates of oral gape position evolution | | |
| | | Rates of relative maxillary length evolution | | |
| | Diversity-dependent speciation | Species diversity | Increasing diversity (e.g., stronger biotic interactions) results in greater specialization and faster speciation rates | 9, 18 |
| Abiotic | Climate-driven | Temperature | Warmer, wetter regions increase evolutionary speed, support more individuals, and hence higher speciation rates | 5, 50 |
| | | Surface runoff | | |
| | Habitat-driven | Area | Larger areas, higher elevations, or greater structural habitat heterogeneity allow the coexistence of more species and enable higher isolation and allopatric speciation | 6, 21 |
| | | Elevation | | |
| | | Soil diversity | | |
| | | Stream gradient | | |

The table provides an overview of general hypotheses for the spatial variation in speciation rates, outlining the main mechanisms, variables used in our study, underlying predictions, and references.

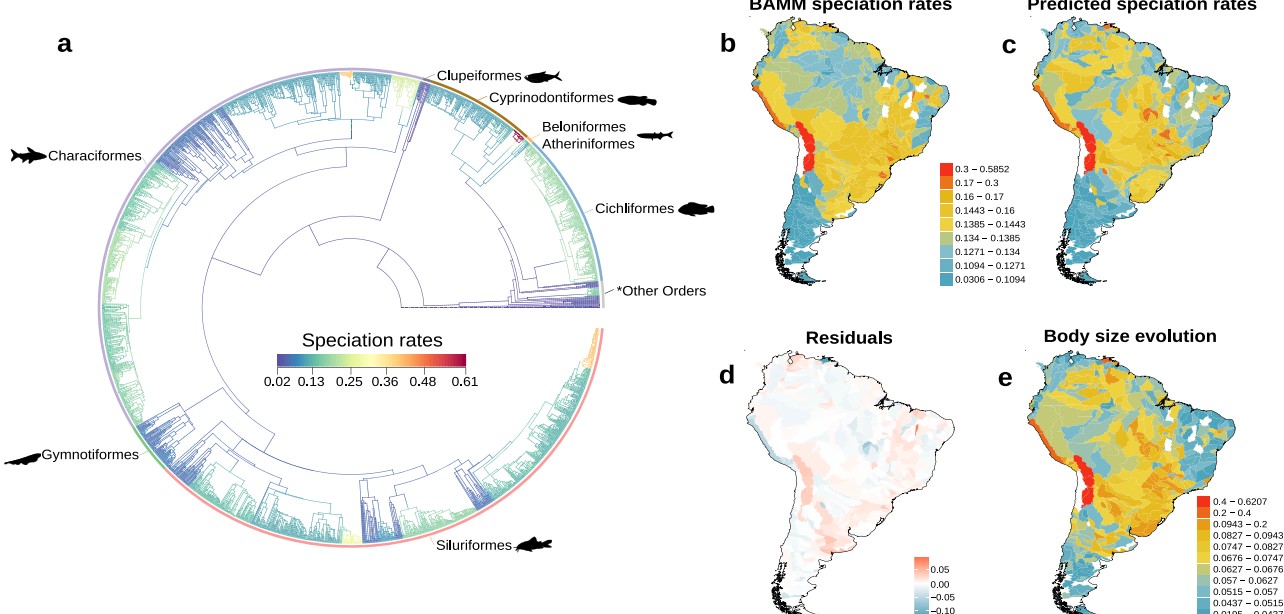

**Fig. 1 | Speciation rates dynamics across phylogeny and space.** Speciation rates were estimated using the Bayesian analysis of macroevolutionary mixtures (BAMM) of 2638 species of South American freshwater fishes. **a** BAMM estimator indicates a wide range of rates among lineages, with branches colored to represent slower speciation rates (blue and green) and faster rates (orange and red). **b** BAMM tip speciation rates averaged for each sub-basin ($n = 460$). **c** Predicted speciation rates from a multiple linear regression including eleven biotic and abiotic factors. **d** Residual spatial variation (red = higher and blue = lower speciation rates than predicted). **e** Rates of body size change averaged for each sub-basin. The color scale in panel **b**, panel **c**, and panel **e** ranges from blue (slower speciation rates) to red (faster speciation rates). Tip speciation estimates in panel **a** were used to map the spatial distribution of speciation rates. "*Other Orders" includes *Perciformes*, *Pleuronectiformes*, *Mugiliformes*, *Synbranchiformes*, *Galaxiformes*, and *Osteoglossiformes*. Silhouettes of *Cichliformes* and *Cyprinodontiformes* were created by Cesar Julian, while that of *Characiformes* was created by Camilo Julián-Caballero and were sourced from phylopic.org. Source data are provided as a Source data file.

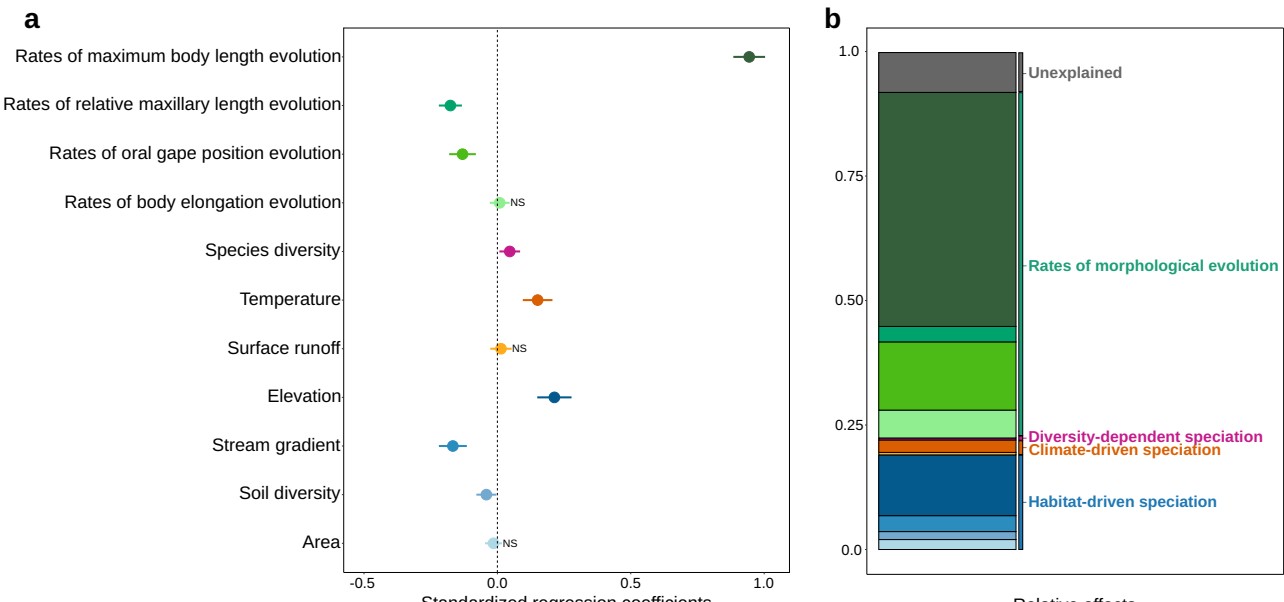

**Fig. 2 | Correlates of speciation rates: biotic (morphological evolution and species diversity) and abiotic factors (climate and habitat characteristics).** **a** Standardized regression coefficients (dots) and their 95% confidence intervals (error bars) are shown for each predictor in the multiple linear regression analysis. **b** Hierarchical partitioning shows the relative importance of each predictor, expressed as the percentage of explained variance. The same color scheme is used in both panels to represent the four main mechanisms. Predictors with statistically non-significant relationships with speciation rates are denoted by 'NS.' The relationship between speciation rates and biotic and abiotic factors is based on sub-basin-level data ($n = 460$). Source data are provided as a Source data file.

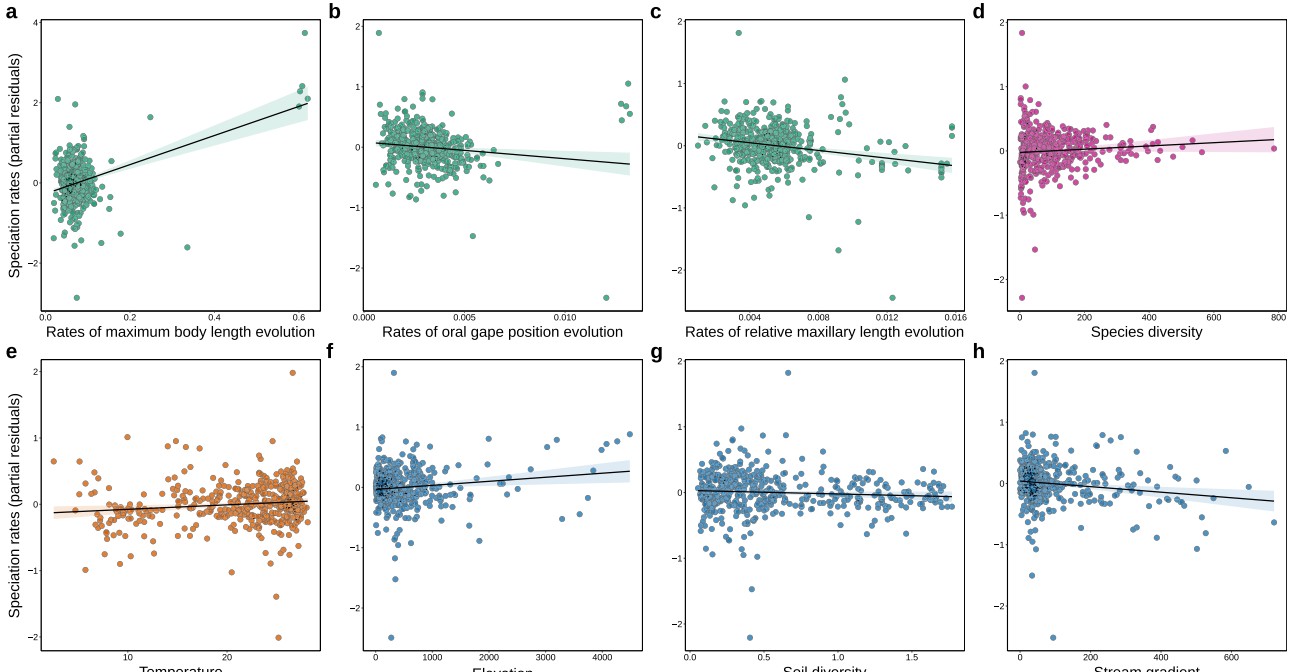

**Fig. 3 | Partial residual plots showing the relationships between speciation rates and biotic and abiotic variables.** These plots were derived from a multiple linear regression analysis of the relationship between speciation rates and **a** rates of maximum body length evolution, **b** rates of oral gape position evolution, **c** rates of relative maxillary length evolution, **d** species diversity, **e** temperature, **f** elevation, **g** soil diversity, **h** stream gradient. The colors indicate the four main classes of mechanisms identified in Table 1: rates of morphological evolution (green), diversity-dependence (pink), climate-driven (orange), and habitat-driven (blue). The relationship between speciation rates and biotic and abiotic factors is based on sub-basin-level data ($n = 460$). Shades around the linear trend line indicate the 95% confidence interval. Notably, there are five outliers clusters in panels **a** and **b**, which correspond to sub-basins predominantly occupied by *Orestias* species. Supplementary Figs. 18 and 19 show qualitatively similar results after removing these outliers. Source data are provided as a Source data file.

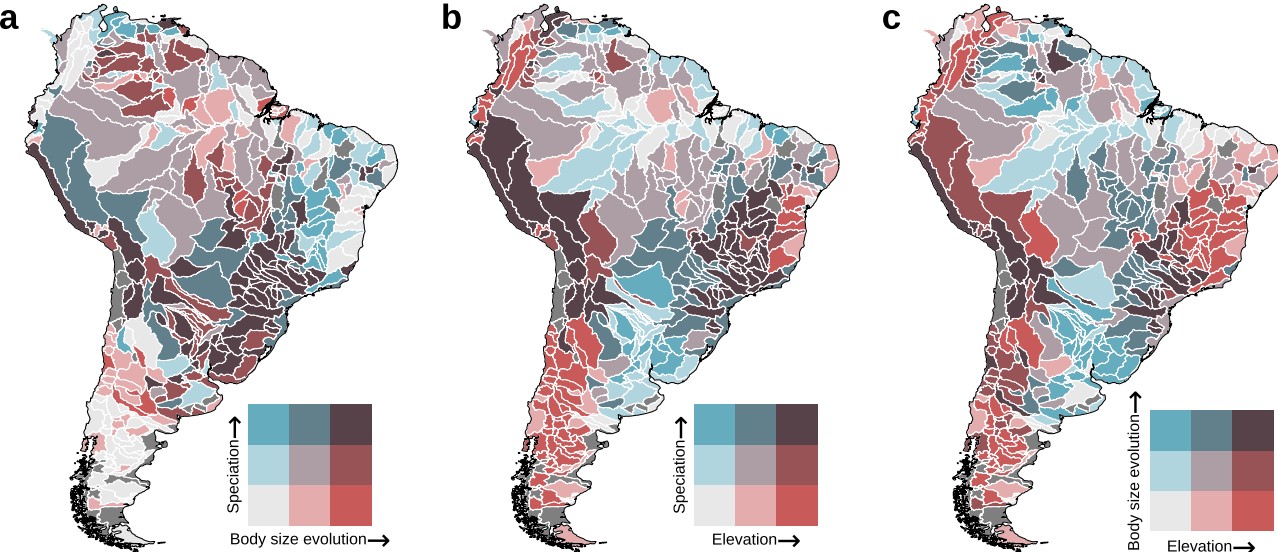

**Fig. 4 | South American regions where speciation rates, body size evolution, and elevation are well-matched or mismatched. a** Spatial covariation between rates of speciation and body size evolution. **b** Spatial covariation between rates of speciation and elevation. **c** Spatial covariation between rates of body size evolution and elevation. Colors correspond to the mean values of speciation rates, body size evolution, and elevation in each sub-basin ($n = 460$). Source data are provided as a Source data file.

spatial variation in speciation rates ($R^2 = 0.470$; Fig. 1e and Fig. 2). This suggests that higher rates of speciation are associated with accelerated body-size evolution (Fig. 3a and Fig. 4a; Supplementary Fig. 2). Evolutionary rates of oral gape position and relative maxillary length exhibited a negative relationship with speciation rates ($R^2 = 0.137$ and $R^2 = 0.031$, respectively; Fig. 2 and Fig. 3b, c). The rate of body

elongation evolution had no significant association with speciation rates (Fig. 2a). These findings are supported by rate-through-time plots, which reveal an acceleration of body size evolution over time, notably around 50 Mya, coinciding with an increase in speciation rates. In contrast, rates for oral gape position and relative maxillary length remained relatively constant or decreased during this period

**Fig. 5 | Variance partitioning analysis shows the unique and joint effects of multiple predictors of speciation rates. a** The amount of variation in speciation rates explained by biotic (rates of morphological evolution and species diversity) and abiotic factors (climate and habitat). **b** The influence of biotic and abiotic factors on speciation rates was broken down into four classes of mechanisms: rates of morphological evolution (green), diversity-dependence (red), climate-driven (orange), and habitat-driven (blue). The variables merged inside each mechanism can be found in Table 1. The numbers in the Venn diagram represent the percentage of explained variance. The sizes of the ellipses are proportional to the amount of variance explained. Source data are provided as a Source data file.

(Supplementary Fig. 3). Additionally, we observed a weak but significant, positive relationship between speciation rates and species diversity ($R^2 = 0.004$; Fig. 2 and Fig. 3d).

Our model also uncovered the considerable importance of abiotic factors on speciation rates ($R^2 = 0.202$; Fig. 2; Supplementary Table 1). The strongest contribution was from habitat-related variables ($R^2 = 0.172$; Fig. 2b), with speciation rates increasing with elevation (second-highest effect size, Fig. 3f and Fig. 4b) but decreasing with soil diversity (Fig. 3g) and stream gradient (Fig. 3h). Contrary to expectations, sub-basins with larger areas did not exhibit higher speciation rates (Fig. 2a). The overall climate-related effects were weak and mainly represented by temperature ($R^2 = 0.025$; Fig. 2), which showed a positive correlation with speciation rates (Fig. 3e). Speciation rates were not correlated with land surface runoff (Fig. 2a).

To examine the unique and joint effects of morphological evolution, diversity-dependence, climate, and habitat on speciation, we performed a variance partitioning analysis (Fig. 5). The results showed that the joint effect of biotic and abiotic factors explained most of the variation in speciation rates (Fig. 5a). Specifically, morphological evolution (biotic) uniquely explained 36% of the variance in speciation rates. However, most of the variation in speciation (52%) could be jointly explained by morphological evolution and habitat variables (Fig. 5b). All other portions of the variation were consistently weak, accounting for less than 6%.

To ensure the robustness of our findings, we addressed a range of technical and methodological artifacts (see Supplementary Note 1: Sensitivity Analyses for a comprehensive discussion). These sensitivity analyses primarily considered uncertainties in estimating rates of speciation and morphological evolution (Supplementary Figs. 4–13), the influence of paleoclimate on speciation rates (Supplementary Figs. 14 and 15), and the potential impact of biological outliers on our results (Supplementary Figs. 16–19). By considering these potential limitations, we can confidently assert that our findings remain robust. They emphasize the importance of body size evolution, particularly its accelerated pace in upland areas, as a key factor in explaining the observed increase in speciation rates.

## Discussion

We evaluated the association of biotic and abiotic factors with spatial variation of recent speciation rates in South American freshwater fishes—the most species-rich continental vertebrate fauna. We found that the rates of morphological evolution, notably body size, are the primary correlates of spatial variation in speciation rates among fishes (Fig. 1e and Fig. 2). We also found that abiotic factors, particularly elevation, appeared to play a role in driving speciation rates, where higher-elevation lineages speciate at faster rates. Further, these two variables were coupled, such that regions in which fish underwent faster body size evolution and

speciation are commonly found at high elevations (Fig. 4). These results are robust, as demonstrated by their consistent performance when tested against several alternative methods and sensitivity analyses (Supplementary Note 1). Taken together, these findings suggest that habitat characteristics can promote speciation by affecting how traits evolve. Therefore, to better understand the origins of present-day biodiversity patterns, it is essential to consider multiple types of traits, their rates of evolution, and the coupled effects with climate and habitat conditions.

The evolutionary rate of one key morphological trait, maximum body length, emerged as the strongest factor associated with differences in speciation rates across regions (Fig. 2). The sub-basins that concentrate species with the fastest speciation rates, such as the Altiplano and Pacific slope of Peru and mesic sub-basins in the Atlantic Forest, also showed an exceptional rate of change in body size (Fig. 1b and Fig. 4a). Particularly, this most recent rapid speciation occurred in clades of small-bodied fishes, such as *Orestias* (Cyprinodontiformes) restricted to high-elevation (>1000 m.a.s.l) lakes and rivers in the Andes[30], and *Hypostomus* (Siluriformes) principally distributed in rivers draining the Brazilian Shield and La Plata basin[31]. While our goal was to evaluate spatial variation in potential drivers of speciation, we also confirmed that a similar outcome unfolded over geological timescales (for more comprehensive coverage, see[25]). For instance, we found that during the Paleogene period (66 Mya), an acceleration in speciation rates coincided with an increase in body size evolution, while rates of other traits declined (Supplementary Fig. 3). These findings provide evidence supporting the Paleogene radiations[32]. Following the Cretaceous-Paleogene mass extinction[32], new species emerged to occupy newly available ecological niches[33], with high-elevation sub-basins potentially playing a crucial role as a speciation hotspot for the modern ichthyofauna during the Neogene[23].

For many vertebrates[7,8] and plants[34], a positive relationship exists between body size evolution and speciation rates. There are several possible explanations for this positive relationship. One possibility is that the evolution of morphological novelty (morphological evolvability) triggers speciation by allowing a clade to diversify and occupy spaces previously inaccessible (adaptive radiation theory[11]). Another possibility is that the speciation process itself leads to rapid morphological change (punctuated equilibrium theory[35]), in which clade evolution is characterized by long periods of evolutionary stasis followed by rapid bursts of morphological change around the time of speciation[36]. Many examples of rapid adaptive radiation are recognized within highly speciose communities[37], such as the close association between body size evolution and elevation identified here. While it is difficult to distinguish between adaptive radiation and punctuated equilibrium theories[36], morphological evolvability emerges as a more plausible and direct explanation for the increased speciation rates.

Rates of speciation and morphological evolution can also be uncoupled[15,16] or even negative[38,39]. The results presented here provide additional evidence to support this possibility, as we found a negative association between speciation rates and rates of oral gape position and relative maxillary length evolution (Fig. 2a and Fig. 3b, c). These findings suggest that speciation is not related to high rates of change in traits associated with feeding behavior, echoing previous studies in snails[40], damselfly[41], lizards[42], salamanders[43], and birds[44]. The inverted relationship between rates of speciation and morphological evolution likely emerged in a non-adaptive scenario, whereby bursts of species formation occur in groups with minimal morphological disparity[45]. These findings suggest that different traits may vary in importance in speciation[13], and considering only one axis of morphological variation, such as body size, may obscure important aspects of the relationship between speciation and morphological evolution[8]. Overall, we argue that considering multiple axes of morphological variation is essential for gaining a complete understanding of the mechanisms driving speciation.

Abiotic factors additionally influenced recent speciation rates, with habitat characteristics explaining more spatial variation in speciation than climate (Fig. 2 and Fig. 5b). Elevation, the best predictor among the abiotic factors, indicates that higher-elevation lineages speciate at faster rates (Fig. 2 and Fig. 4b). There are many possible explanations for why elevation can affect speciation rates[6]. Areas formed during active uplift have more geographic barriers, physiographic heterogeneity, and new habitats[6,46], all of which can lead to reproductive isolation[3,47] and ultimately spur speciation. Our results are consistent with many cross-taxa observations that demonstrate faster speciation in mountains[46,48], although the link between speciation and morphological evolution along elevational gradients remains largely unexplored. We observed that body size evolves faster in high elevations (Fig. 4c), which in turn correlates with a higher speciation rate. Body size is a fine-tuned trait directly linked to nearly every aspect of organismal biology (e.g., habitat use, life history, and metabolism[10]), and changes in body size are associated with ecological differentiation and reproductive isolation[10]. Therefore, environmental conditions along elevation gradients are likely to have imposed evolutionary pressures on body size that are tightly coupled with speciation rates.

Other factors played little or negligible roles in the rates of fish speciation across South American sub-basins. Proxies of habitat conditions, namely stream gradient, and diversity of soil types, had a weak negative relationship with speciation rates (Fig. 2 and Fig. 3g, h), indicating that the highest speciation rates occur in regions with lower soil diversity than Amazonian lowlands and lower gradients. These patterns are atypical, as they are contrary to the usual increase in speciation rates with greater environmental heterogeneity and a number of available niches[49]. The area was found to be unimportant, suggesting that increases in sub-basin size are not sufficient to drive a significant increase in speciation rates (Fig. 2). The effects of species diversity were also minimal (Fig. 2), although we did find higher speciation rates in areas with higher species richness. Greater species diversity itself can promote speciation by increasing the probability of divergence between populations, structural complexity, or through a trade-off between competition and dispersal[9,18] (but see[17] for a contrary view). Finally, we found that temperature was positively related to speciation rates (Fig. 3e). These results support literature stating that warmer and productive regions increase demographic rates, metabolic rates, and evolutionary rates[50], yet a recent wave of studies has shown the fastest rates of speciation in cold regions[51].

Speciation rates vary by several orders of magnitude across regions[51], but a comprehensive view of their causes and consequences continues to puzzle scientists[4]. To date, much of what we know about drivers of speciation comes from systems at smaller spatial and phylogenetic scales than our study or does not evaluate multiple factors simultaneously. Our study is notable for its scale, incorporating a vast dataset of the most diverse vertebrate groups worldwide and considering multiple potential drivers of spatial patterns in speciation. We revealed that the rate of morphological evolution is the strongest predictor of geographic variation in recent speciation rates, supporting a positive relationship between rates of body size change and speciation rates[7,8]. Our findings also support the idea that changes in body size are likely to be accelerated in upland regions, leading to an increase in speciation rates of living fishes. Moreover, we found that elevation has a more significant relationship with rapid speciation than do climate, area, and diversity-dependent mechanisms. By evaluating a baseline set of biotic vs. abiotic factors, our work suggests a synergy between morphological evolution and habitat characteristics in shaping large-scale patterns in speciation.

## Methods

### Geographic occurrences and phylogenetic data
We obtained presence/absence data for 4967 South American species across 460 sub-basins[25] delimited by the HydroBASINS framework (level 5[52]). This dataset is based on an extensive survey of web repositories (e.g., GBIF, specieslink) and literature sources[25], with several procedures to increase data quality (e.g., removal of georeferencing errors, exotic and migratory species). In addition, simulation approaches were performed to reduce sampling effort heterogeneities in species distribution[25] by calculating a completeness index in each drainage basin based on the probability that the next sampled record would add a new species to those observed in the focal basin.

We used a newly compiled, time-calibrated tree of Neotropical Freshwater Fishes[25]. The tree was built on 5984 terminal taxa, including 3169 species with available genetic data (51 independently aligned and trimmed markers), 31 fossil-constrained nodes, and 2815 species inserted by taxonomic imputation. This tree summarizes the current knowledge on phylogenetic relationships among Neotropical freshwater fishes.

### Estimating speciation rates and assessing their reliability
To explore diversification patterns among lineages, we used Bayesian Analysis of Macroevolutionary Mixtures[53] (BAMM), diversification rate (DR) statistic[54], and Missing State Speciation and Extinction (MiSSE) metric[55]. We employed species-specific diversification rate (tip-based estimates), which are more accurate than deep-time estimates[55,56]. BAMM allows the modeling of complex speciation and extinction dynamics by detecting major shifts in evolutionary rates from a time-calibrated phylogeny[53,57]. We ran 20 million generations of Markov Chain Monte Carlo (MCMC) over four chains, sampling every 2000 generations and accounting for incomplete taxon sampling (global sampling fraction of 0.53). We discarded the first 25% of the MCMC samples as burn-in and ensured that the effective sample size of all parameters was above 200 (CODA R package[58]). The prior settings for speciation and extinction were obtained by the *setBAMMpriors* function in the BAMMtools R package[59]. The DR statistic is calculated as the inverse equal splits rates (i.e., the sum of the branch lengths separating a tip from the root) and has been shown to reflect speciation rates better than net diversification rate[54,57]. Finally, MiSSE is a trait-free version of the HiSSE framework, designed to accurately estimate tip diversification rate while accommodating a broad range of speciation and extinction scenarios[55].

While our model estimated both speciation and extinction rates, we focused on reporting the speciation estimates, as inferring extinction rates from extant phylogenies is known to be biologically unrealistic[60,61]. Therefore, we extracted tip-based (recent time) speciation rates from BAMM, DR, and MiSSE and explored spatial patterns by averaging these rates among co-occurring species in each sub-drainage basin. While the use of mean values is a common practice, we recognize that it may not always be suitable, especially when analyzing communities with high levels of variation in the variable of interest. To

address this issue, we calculated the median, minimum, and maximum values of speciation rates across sub-basins and assessed the impact of predictor variables on all of these summary statistics. Our analysis revealed that spatial patterns of speciation were largely consistent when using mean, median, minimum, or maximum values (Supplementary Fig. 20). This was particularly evident for regions with lower (e.g., Amazon basin) and higher (e.g., upland sub-basins) rates of speciation. We also found that the importance of the predictors remained qualitatively similar across different summary statistics (Supplementary Fig. 21), further highlighting the major role of body size evolution in driving speciation.

We found that speciation rates estimated from alternative methods were strongly correlated (Supplementary Fig. 4), indicating that these methods are likely to produce consistent results (Supplementary Fig. 5). Therefore, we focus further analyses on BAMM speciation rates (see[8,57] for method comparison) and the results using DR and MiSSE estimates are available in the supplementary material. BAMM speciation estimates were highly correlated with net diversification rate (BAMM speciation minus BAMM extinction) (Supplementary Fig. 6), suggesting that our findings are likely to hold for diversification rate as well. We used the most up-to-date and comprehensive reconstruction of the evolutionary relationships among South American fish species[25]. Nonetheless, alternative phylogenies exist, and BAMM speciation estimates could vary across phylogenies and affect our interpretation (Supplementary Note 1). We, therefore, compared our estimates with (i) the corresponding tree based solely on genetic data[25], (ii) a ray-finned fish supertree[62], or (iii) taxon-specific trees such as those for poeciliidae[63], cichlid[26], and characoid[24] fishes. We obtained similar BAMM speciation estimates across phylogenies (Supplementary Fig. 7).

## Morphological traits, their rates of evolution, and robustness

We selected five morphological traits from the Fishmorph database[64] to capture relevant ecological, physiological, and behavioral dimensions of fish species: body elongation, relative eye size, oral gape position, relative maxillary length, and maximum body length. Body elongation, functionally related to swimming performance and habitat utilization, has been identified as a dominant axis of phenotypic disparity in many clades[65,66]. Relative eye size is related to visual acuity and diel activity patterns[67,68]. Oral gape position and relative maxillary length are related to feeding behaviors and trophic position[69,70]. Maximum body length has been correlated to many important evolutionary processes, like metabolic rate, mutation rate, generation time, organismal vagility, species geographic range, and population structure[71,72]. Body size estimates were $\log_{10}$-transformed prior to estimating evolutionary rates, following the suggestion for size data[73].

We pruned the phylogeny of Neotropical freshwater fishes[25] to match the South American species with morphological data, resulting in 2638 species with phylogenetic information, geographic occurrences, and morphological data (53% of the South American ichthyofauna). Species missing morphological data are not phylogenetically clustered (Supplementary Figs. 22-26; tested with *miss.phylo.d* function[74]), therefore, our results do not over or under-represent any particular fish clade. As trait values were unknown for a small number of species in the phylogeny (0.6–10% among the five traits; Supplementary Figs. 22–26), we imputed missing trait data using the *Rphylopars* function[75]. This approach uses phylogenetic information for imputation criteria[75] and has been shown to outperform other approaches[76].

Posteriorly, we estimated per-lineage rates of morphological evolution of the five traits by using the BAMM 'trait' module[59]. We ran three independent MCMC chains with 100 million generations each, sampling every 10,000 generations and removing the first 25% of samples as burn-in. We ensured that the effective sample size was above 200. Based on tip-based evolutionary rates, we calculated the

mean rates of morphological evolution for each sub-basin using their corresponding species assemblages (Supplementary Figs. 27–31).

We further compared the trait evolution estimates from the BAMM trait model with another robust method, known as the BayesTrait[77] metric (settings are described in Supplementary Note 2: Extended Methods). Our analyses revealed no evidence of bias in the patterns of morphological evolution that could be attributed to the choice of method, as the estimates of morphological evolution obtained using both BAMM and BayesTraits methods exhibited strong consistency (Supplementary Figs. 8–12). As the proportion of sampled species was unequal among genera (Supplementary Table 2) and an incomplete taxonomic sampling may influence estimates of evolutionary rates[78], we also verified the sensitivity of our estimates to a higher species coverage. More precisely, we retrieved data from Fishbase[79] on maximum body length and reanalyzed the correlation between the rate of speciation and body size evolution with 4228 species (85% of the South American ichthyofauna). We showed that an increased taxonomic sampling produced spatial patterns of speciation and body size evolution consistent with our pruned data set (Supplementary Fig. 13).

## Species diversity

We determined species diversity as the total number of fish species co-occurring in each subdrainage basin (Supplementary Fig. 32). Considering species richness and area as predictors in the model assumes a slope of the species-area regression equal to one, although this scaling exponent has been shown to empirically range from 0.25-0.50[80]. To address this issue, we reanalyzed the patterns of species richness in the context of species-area relationship[80], following the power function: $SD = SR/A^z$, where SD is the species density (i.e., number of species per unit area), SR is the number of species in area $A$, and $z$ is the species-area scaling exponent (i.e., the slope of the species-area regression). Overall, we found that species density and area predicted speciation rates similarly to when using raw species richness (Supplementary Figs. 33 and 34), and, therefore, we provide these results as supplementary.

We conducted additional analyses to assess the potential impact of the number of species and wide-ranging species on our conclusions by: (i) excluding all sub-basins with less than 10, 15, and 20 species and (ii) applying a weighting approach to account for the disproportionate influence of wide-ranging species on the mean estimates across sub-basins. The similarity of results from these analyses indicates that our findings are robust (Supplementary Figs. 35 and 36), unaffected by potential biases related to the number of species and the inclusion of wide-ranging species.

## Climate variables

We obtained data on four variables directly related to climate: (i) annual mean temperature, (ii) annual mean precipitation, (iii) actual evapotranspiration, and (iv) land surface runoff. Annual mean temperature and precipitation were taken from the WorldClim Version 1 database at 2.5 arc-min resolution[81] (Supplementary Table 3). Air temperature was used because it is strongly associated with water temperature[82], which is not available for most of the sub-basins. Data on actual evapotranspiration and land surface runoff were extracted from the HydroATLAS database at 15 arc-s resolution[83]. We calculated the mean values of all the variables for each of the 460 sub-basins (Supplementary Fig. 37).

As past climate may leave imprints on present-day speciation rates, we examined the potential effect that past temperature had on speciation rates. Past temperature data corresponding to the Pliocene (ca. 3.3 My) and Last Glacial Maximum (LGM; ca. 21 ka) were extracted from the CHELSA database (at 2.5 arc-min resolution)[84]. Overall, past temperature explained speciation rates in a similar fashion to contemporary temperature and was treated as supplementary

(Supplementary Figs. 14). We also explored the paleoclimate effects by employing a temperature-dependent speciation model[85]. This model incorporated temperature variation through the Cenozoic epoch[86] (~67 Mya) and only clades with more than 50 species originating within this time frame (Supplementary Note 2). The results of this analysis reaffirmed our findings, indicating that 73% of the examined clades demonstrated a positive effect of temperature on speciation (Supplementary Fig. 15).

### Habitat variables

To represent the habitat, we considered: (i) mean elevation, (ii) geographic area of the sub-basin (measured in square meters), (iii) stream gradient, and (iv) soil diversity. We also tested the impact of alternative measures related to elevation (i.e., topographic complexity), which included standard deviation in elevation, terrain slope, and basin relief. Importantly, mean elevation was a better predictor of speciation rates, and therefore, we considered topographic complexity measures as supplementary (Supplementary Fig. 38).

Data on elevation were taken from the WorldClim Version 1 database at 2.5 arc-min resolution[81]. The geographic area of the sub-basin was obtained from HydroBASINS (level 5[52]) and stream gradient and soil types from HydroATLAS at 15 arc-second resolution[83] (Supplementary Table 3). Soil diversity was measured using the Shannon diversity index based on substrate types and soil conditions within each sub-basin. We calculated the mean values of all the variables for each of the 460 sub-basins (Supplementary Fig. 39).

### Statistical analysis

To examine the connections between geographic variation in recent speciation rates and various factors, including biotic factors (such as rates of trait evolution and species diversity) as well as abiotic factors (such as climate and habitat variables), we performed a multiple linear regression analysis. Prior to the regression analysis, we assessed multicollinearity among predictors by examining the variance inflation factor[87] (VIF). Three variables with a high VIF were removed from the analyses (rate of evolution of relative eye size, annual precipitation, and actual evapotranspiration; Supplementary Fig. 40). In this way; we retained eleven variables with low multicollinearity (VIF < 5; Supplementary Table 1). To make the effect size comparable among predictors, we standardized response and predictor variables using z-score standardization (mean = 0; sd = 1), with the sub-basin area being log-transformed before z-score standardization.

To quantify the relative contribution of each predictor in explaining the total variance in speciation rates, we performed hierarchical partitioning (metric *lmg*) as implemented in the R package relaimpo[88]. This procedure decomposes the model-explained variance into non-negative contributions and evaluates the relative importance of each predictor variable in linear models[88]. We also used variance partitioning to investigate the unique and shared effects of four major mechanisms as predictors of speciation rates. The predictor variables were consistent with those used in the linear multiple regression and were grouped into (1) rates of morphological evolution, (2) diversity-dependent speciation, (3) climate-driven speciation, and (4) habitat-driven speciation (Table 1).

We further investigated the sensitivity of our results to spatial autocorrelation in the residuals and outliers. To examine spatial autocorrelation, we computed Moran's $I$[89] based on residual speciation rates (i.e., the residuals from a linear multiple regression of the 11 predictors on speciation rates). We found low spatial autocorrelation in the residuals from the non-spatial, multiple regression (Global Moran's $I$ = −0.003, $P$ = 0.423; Supplementary Fig. 41). Finally, the genus *Orestias* (Cyprinodontiformes) is endemic to the Andean basin of Peru, Bolivia, and Chile and had a recent and localized burst in diversification[30], which was also recovered in our speciation estimates (Fig. 1a). To confirm that our primary conclusions were not influenced

by *Orestias* species, we (i) ran the BAMM model after removing all *Orestias* species (ii) excluded *Orestias* species from the sub-basins where they are present, and (iii) removed five sub-basins predominantly occupied by *Orestias* species and with exceptional rates of speciation. The removal scenarios involving *Orestias* had no impact on our ability to detect high rates within other clades (Supplementary Fig. 16) or introduce bias to the mean speciation rates across sub-basins (Supplementary Figs. 16–19).

### Reporting summary

Further information on research design is available in the Nature Portfolio Reporting Summary linked to this article.

## Data availability

Fish species occurrence records and phylogeny are from Cassemiro et al.[25]. Morphological traits are available at FISHMORPH database (Brosse et al.[64]). Climate and habitat data are from WorldClim (www. worldclim.org) and HydroATLAS database (www.hydrosheds.org/hydroatlas). All data supporting the findings of this study are available in the Zenodo database under Attribution 4.0 International license and accession code: https://doi.org/10.5281/zenodo.8301082. Source data are provided in this paper.

## Code availability

The R codes employed for the analyses undertaken in this study can be accessed in the Zenodo database under the Attribution 4.0 International license. The corresponding accession code is https://doi.org/10.5281/zenodo.8301082.

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

## Acknowledgements

This study was supported in part by the Coordenação de Aperfeiçoamento de Pessoa de Nível Superior—Brasil (CAPES)—Finance Code 001; the Swiss Federal Institute for Forest, Snow and Landscape (WSL); and the National Institutes for Science and Technology (INCT) in Ecology, Evolution and Biodiversity Conservation, supported by MCTIC/CNPq (proc. 465610/2014-5) and FAPEG (proc. 201810267000023). F.O.C. was supported by a doctorate and a 'sandwich' fellowship from CAPES. C.H.G. and E.B. acknowledge funding support from the European Research Council (ERC) under the European Union's Horizon 2020 research and innovation program (grant 787638) and the Swiss National Science Foundation (grant 173342). M.T.P.C. and C.H.G. are financially supported by the Swiss National Science Foundation (SNSF, no. 315230_197753). We also thank the Graham group for the insightful discussions on the paper.

## Author contributions

F.O.C., C.S.D and C.G. designed the study. F.O.C. collected and analyzed the data with the crucial assistance of M.T.P.C., E.B., F.A.S.C. and J.A. All authors (F.O.C., C.S.D., M.T.P.C., F.A.S.C., E.B., J.S.A., R.O.W., C.H.G.) actively participated in the interpretation and discussion of the results, providing valuable insights throughout the drafting process.

## Competing interests

The authors declare no competing interests.
