## [Peer Review File · Nature Communications]

REVIEWER COMMENTS

Reviewer #1 (Remarks to the Author):

The study by Cerezer et al. addresses the question of what factors contribute to the exceptional diversity of Neotropical freshwater fishes. The authors utilized a comprehensive dataset of species geographic distributions, phylogenetic, morphological, climatic, and habitat data, providing a wealth of information for their analysis. They found that recent speciation events are strongly associated with body size evolution, particularly in clades with small adult body sizes and in clades that inhabit higher elevations. Their results also highlight the negligible effects of several factors that are widely thought to facilitate speciation, including temperature, area, diversity-dependence, and even latitude/longitude. While their findings are of broad interest to the Nature Communications readership, there are some major issues that need to be addressed.

My primary concern with the study is its limited consideration of the temporal dimension in evolution. Although the authors conduct some analysis of paleotemperature, a more comprehensive examination of the variation in speciation rates over time would significantly enhance the study. Focusing mostly on tip rates and present-day abiotic factors ignores an essential factor in evolution: time. The overarching goal of the paper, "Here we measure the degree to which biotic and abiotic factors have shaped recent variation in speciation rates across space," would be more comprehensive if time were also included (i.e., "...across space and through time"). Investigating how biotic and abiotic factors have influenced speciation rates over different geological periods can provide insights into the historical and ongoing factors that shape biodiversity patterns in Neotropical fishes.

I am unclear on the analysis conducted by the authors on Figs. S16 and S17, which examines the relationship between paleotemperatures and speciation rates. A more direct test of the relationship between these two variables would be to plot them in the same graph (e.g., lineage-through-time plot + paleotemperature plot). Additionally, the environmental-dependent diversification model implemented in R-Panda can be used to identify a relationship more directly between these variables. It would also make sense to examine the association between morphology and paleoclimate data using an OU climate model (e.g., <https://www.pnas.org/doi/full/10.1073/pnas.2122486119>).

The authors have cited the paper <https://onlinelibrary.wiley.com/doi/full/10.1111/evo.14517>, but have not utilized MiSSE, the method proposed in that paper. As the results of diversification rate estimation can vary significantly depending on the method used (this is acknowledged by the authors), it is crucial to compare more than just two metrics. MiSSE could serve as a valuable third option as it functions as an intermediate between BAMM and ClaDS, thereby becoming a more flexible model-based approach that can identify areas of rate heterogeneity or homogeneity at the tips of the tree. For these same reasons, I'm not comfortable with this premise: "we focus further analyses on BAMM speciation rates." I think results from all analyses need to be reported in tandem.

Finally, I would like to applaud the authors for dedicating a section of the Results to account for uncertainty and sensitivity (section 3.3). While this is a great initiative, I believe this section could benefit from a more thorough interpretation of the evolutionary results in light of this uncertainty. This is particularly relevant since some trees produce highly divergent BAMM results (Figure S2): e.g., "the ray-finned fish supertree of 57, ρ ranged from 0.61 to 0.64; Supplementary Fig. 2b-c."

Regarding the morphological analyses, the use of a single approach (BAMM 'trait') may also be overly simplistic. Comparing the BAMM results with BayesTraits v4, which incorporates rate heterogeneous models, would provide a more comprehensive analysis.

In section 2.5, the authors tested for associations among recent speciation rates and biotic and abiotic factors using multiple linear regression. However, it does not appear that they used phylogenetic regression analyses. It's important to note that even when phylogeny-based tip rates are used, an OLS regression analysis does not account for phylogenetic non-independence. This is

a common mistake. I'm not sure how to perform a phylogenetically-corrected multi-linear regression analysis, but I believe there are options available. For linear regression analyses, I use `phylolm` and start by comparing OLS (no phylogeny or star phylogeny) vs. PGLS-BM (Brownian motion) vs. PGLS-OU (OU) models to find the best-fit model. Then, I report only regression analyses based on the best-fit models (sometimes a poor-fit model may produce a better p-value, but that's just an artifact due to model misspecification). It's crucial to follow all these steps for each regression analysis conducted, as it's not safe to assume that the best-fit model for one regression is also the best-fit model for another regression. Find a way to apply the same rationale to phylogenetically-corrected multi-linear regression analyses.

Minor comments:

"2.3 | Rates of morphological evolution and species diversity." This subsection covers three different topics: the collection of morphological traits, the estimation of rates of morphological evolution, and the species diversity per basin/species area relationship. Consider splitting it into multiple subsections to improve the flow. In fact, the last paragraph on species diversity seems more connected with the previous section, as it is related more to speciation rates than to morphology: "overall, we found that species density and area predicted speciation rates similarly."

"2.4 | Abiotic variables." Same thing here. The title of the subsection leaves the impression that it is only about collection of abiotic variables. But there's also analyses linking abiotic variables with speciation rates. Perhaps just add one subsection linking biotic (morphology) and abiotic factors with speciation rates.

I feel that the M&Ms and Results sections in the main text are too condensed. The supplement would benefit from an expanded explanation of the research approach and findings in these sections.

Typo: "in which are more accurate than deep-time estimates."

Figs. S16–S18. Do discuss something about the outliers in these plots. Also, why report r rather than r^2 values? And p values?

The maps displaying rates of evolution are some of the most striking results of this paper. It would be beneficial to include a map with rates of body size evolution in the main text. This could be achieved by adding a new figure or panel to Figure 1.

The figures in the PDF I downloaded in the main text (but not the SM file) have extremely low resolution (cannot read axis labels in many plots). Make sure high-resolution figures are uploaded for the final version.

The trees plotted on the SM file need taxonomic annotations.

"On the contrary, evolutionary rates of oral gape position and relative maxillary length were negatively related to speciation rates (Fig. 2 and Figs. 3b-c; Table 2). Body elongation evolution had no significant association with speciation rates (Fig. 2a; Table 2). We also observed a weak, but significant, positive relationship between speciation rates and species diversity (Fig. 2 and Fig. 3d; Table 2)." Add R^2 values within the parentheses.

Label all supplementary figures as "Fig. SX" or "Supplementary Fig. X." Same for tables.

Reviewer #2 (Remarks to the Author):

The authors have aggregated an impressive dataset from relevant databases and used a newly published tree to investigate ecological, geographic, and environmental drivers of speciation rates in diverse vertebrate radiation. The taxonomic scale of the analyses is impressive, and the authors

include a robust set of sensitivity analyses to support their methods and results. The finding that high elevation and rapid body size evolution drive high speciation is novel and of broad interest to evolutionary biologists. However, I have identified some weaknesses and flaws that make it hard to determine whether their findings are accurate. Therefore I recommend major revisions to this manuscript.

While BAMM is appropriate for their analyses, the estimates of tip speciation rates across ~2300 species raise some concerns. First, the rate shifts identified by BAMM are very clade-specific and do not appear to recover known rate shifts within clades from other published studies. These include Cichliformes; Geophagini, and Heroini (see Burress & Tan 2017), and Characidae and Anostomidae (see Melo et al. 2022). The authors should compare the estimates of speciation rates in this study with the above-referenced studies as they did for the Poecillidae tree (line 137). At the very least, they need to address why well-known lowland lineages that show elevated speciation rates in other studies using BAMM do not show high rates in their analysis.

Secondly, BAMM speciation rates appear to vary slightly across most of the phylogeny, and exceptionally high rates are concentrated in a few small clades. I am concerned that the exceptional rate of these small clades, particularly Orestias, may impede the ability to detect rate shifts of a smaller magnitude within other clades. Therefore, I recommend removing all Orestias species and repeating the BAMM analyses. Also, it would be helpful for the authors to report the distribution of tip rates in the supplementary materials.

Similarly, in Supplementary Figure 2D, the speciation rates calculated from the Reznick et al. 2017 tree vary on a scale much greater than those calculated in this study, again suggesting that the BAMM analyses may be underestimating rate variation within clades.

Burress, Edward D., and Milton Tan. "Ecological opportunity alters the timing and shape of adaptive radiation." *Evolution* 71.11 (2017): 2650-2660.

Melo, Bruno F., et al. "Accelerated diversification explains the exceptional species richness of tropical characoid fishes." *Systematic Biology* 71.1 (2022): 78-92.

The authors take the mean speciation rate of all co-occurring species within each sub-basin, which I believe is problematic because of the extreme variation in rate between lineages. They should also report the range or distribution of rates within each sub-basin. Because rates appear to vary little across most of the tree (excluding a few outlier clades), including even one or two exceptional species may be enough to inflate the mean rate of a sub-basin with low species richness. Reporting the min and the max rates would at least show whether the overall distribution of rates varied across basins. Also, I suggest performing the multiple regression analysis across the min and max values.

Similarly, the authors did not account for the geographic range of species. Wide-ranging species with exceptional speciation (either high or low) rates may bias estimates. The authors did remove sub-basins with <5, <10, <15, and <20 species; however, they do not report on the significance of these results and make only qualitative comparisons (line 329). Similarly, the authors remove the 5 outlier sub-basins containing exclusively Orestias species; however, it is unclear whether Orestias species occur in other sub-basins and if the authors removed those. Because Orestias rates are exceptionally high, they may still bias the mean speciation rate in any sub-basin they are in, even if that basin is relatively speciose. If the authors did remove all sub-basins with Orestias species before repeating the multiple linear regression (Supplementary Table 3), they should clarify that. In any case, I suggest identifying all sub-basins in which Orestias species occur and removing Orestias from all the sub-basins.

Lastly, the color scale in Figure 1 B and C is misleading. The range of the highest category (red) is greater than the range of all other categories. The lower end of the highest category (~0.18) may not significantly differ from most other basins (0.12-0.18). In contrast, the high range of 0.61 is significantly higher than the lower range of the same category. This color scheme makes it look

like all red basins have a high speciation rate when they might only be .01 different from another sub-basin. The authors should redo the figure with more equal ranges between colors.

RESPONSE TO REVIEWERS' COMMENTS

Dear Reviewers,

Thank you for the opportunity to revise and resubmit our manuscript entitled "Accelerated body size evolution in upland environments drives recent speciation in South American freshwater fishes" (ID: NCOMMS-23-04757). We appreciate the thoughtful comments and feedback provided during this round of the review, which have helped us to improve the quality of our work.

We have carefully incorporated all of the suggestions into the revised manuscript. In particular, we have made significant revisions in the following parts:

1. As an alternative to BAMM, we have employed additional models to estimate speciation rates (DR^1 and $MiSSE^2$) and morphological evolution ($BayesTraits^3$). These additional analyses further support our main findings.
2. In order to strengthen the robustness of our findings, we conducted several additional supplementary analyses. These analyses assessed the estimates of the evolutionary rate, identified and removed outliers, weighted by the species range size, and evaluated additional summary statistics (median, min, and max). We demonstrate that different analytical approaches applied to our datasets consistently lead to the same conclusion: body size evolution is the strongest predictor, and the rate of body size evolution increases with elevation, thereby accelerating speciation even more at high elevations.

3. We have included a discussion on how evolutionary rates change over time. Our findings reveal a consistent trend over time, wherein the rates of speciation and body size evolution exhibit a close association. Remarkably, other traits remained relatively stable or even decreased during periods of accelerated speciation.
4. To investigate the influence of historical temperature on speciation, we have conducted direct tests using a temperature-dependent speciation model. Our findings reveal a positive correlation between historical temperature and speciation for most fish clades, consistent with the spatial patterns observed.

In addition to these major revisions, we have also made minor edits to improve the phrasing, correct typos, and refine the figures and tables. We feel that these changes have significantly strengthened the clarity and rigor of our study. Our main conclusions remained the same, emphasizing the crucial role of body size evolution and its accelerated changes in upland regions as drivers of speciation rates.

To facilitate your review of the revised manuscript, we have included line-to-line comments and changes highlighted in blue in the main text. We have carefully considered all of your suggestions and hope that you will find the revisions satisfactory. Once again, we appreciate the opportunity to submit our work to your esteemed journal and thank you for your consideration. We believe that our revised manuscript will make a valuable contribution to the field and look forward to your feedback.

Sincerely,

Felipe O. Cerezer (on behalf of all co-authors)

Comments from reviewer 1 and respective replies or actions taken:

(1) Reviewer 1: *The study by Cerezer et al. addresses the question of what factors contribute to the exceptional diversity of Neotropical freshwater fishes. The authors utilized a comprehensive dataset of species geographic distributions, phylogenetic, morphological, climatic, and habitat data, providing a wealth of information for their analysis. They found that recent speciation events are strongly associated with body size evolution, particularly in clades with small adult body sizes and in clades that inhabit higher elevations. Their results also highlight the negligible effects of several factors that are widely thought to facilitate speciation, including temperature, area, diversity-dependence, and even latitude/longitude. While their findings are of broad interest to the Nature Communications readership, there are some major issues that need to be addressed.*

Reply: Thank you for reviewing our manuscript and for providing us with such valuable feedback. We have carefully considered all of your comments, and we are pleased to respond to them in this letter. We hope that the revised version has addressed all your concerns and that you will find the paper stronger and more compelling.

(2) Reviewer 1: *My primary concern with the study is its limited consideration of the temporal dimension in evolution. Although the authors conduct some analysis of paleotemperature, a more comprehensive examination of the variation in speciation rates over time would significantly enhance the study. Focusing mostly on tip rates and present-day abiotic factors ignores an essential factor in evolution: time. The overarching goal of the paper, "Here we measure the degree to which biotic and abiotic factors have shaped recent variation in speciation rates across space," would be more comprehensive if time were also included (i.e., "...across space and through time"). Investigating how biotic and abiotic factors have influenced speciation rates over different geological periods can provide insights into the historical and ongoing factors that shape biodiversity patterns in Neotropical fishes.*

Reply: Our study aims to improve understanding of how biotic and abiotic factors affect speciation in Neotropical fishes by focusing on recent events over geography. To this end, we simultaneously evaluated a series of hypotheses in order to obtain a rather comprehensive understanding of recent speciation in the group. Most of the predictor variables (with the exception of temperature data are available from a relatively deep geological perspective) that are central to our main question and hypothesis (sub-basin area, elevation, soil diversity, species diversity) lack paleo reconstructions, preventing us from taking a deep time perspective. Assessing variation in speciation rates over deeper time intervals is indeed an important long-term goal in the study of the target biota, however, it was not possible within our study's scope as our goal was to evaluate the relative importance of multiple different hypotheses in the generation of recent diversity.

Despite these challenges, we have incorporated some deeper time explanations where possible to enrich our study as suggested by the reviewer. First, we tested the relationship between speciation and paleotemperature by running a temperature-dependent speciation model, as suggested in a comment below (Page 12, Line 420 to 426). Additionally, we created rate-through-time plots for both rates of speciation and morphological evolution, allowing for evaluation of how evolutionary rates vary over time (Page 4, Line 121 to 130; Page 6, Line 182 to 190). For example, when plotting speciation rates over time, we observe a steady rate of speciation until the Paleogene, where there is a noticeable increase. Likewise, when examining the rate of trait evolution, we can identify some episodic events that reflect an acceleration or deceleration in the evolutionary process. These findings support our results from a spatial perspective. We briefly mentioned these new perspectives in our results (Page 4, Line 121 to 130) and discussion (Page 6, Line 182 to 190), and all of the rates-through-time plots can be found in the supplementary material (Supplementary Figs. 1 and 3).

(3) Reviewer 1: *I am unclear on the analysis conducted by the authors on Figs. S16 and S17, which examines the relationship between paleotemperatures and speciation rates. A more direct test of the relationship between these two variables would be to plot them in the same graph (e.g., lineage-through-time plot + paleotemperature plot). Additionally, the environmental-dependent diversification model implemented in R-Panda can be used to identify a relationship more directly between these variables. It would also make sense to examine the association between morphology and paleoclimate data using an*

OU *climate* *model* (e.g.,
<https://www.pnas.org/doi/full/10.1073/pnas.2122486119>).

Reply: We agree that the previous correlative analyses between speciation rates and paleotemperature data obtained from CHELSA⁵ could not provide a direct test on how speciation varies with historic temperature. Thus, we follow your recommendation to run a model-based phylogenetic comparative method⁶. We ran a temperature-dependent speciation model to establish a closer association between these two variables (see Supplementary Extended Sensitivity analyses). Due to the rarity and imprecision of historical temperature data for the timeframe of our phylogeny (250 mya), we used paleotemperature data covering a 67 million-year timeframe⁷. As a result, we only evaluated clades that are less than 67 million years old and have a minimum of 50 species, resulting in a total of eleven subclades (Supplementary Fig. 15). We ran the RPANDA model separately for each subclade. This species-level approach recovered a positive association between speciation rates and temperature, similar to that observed in our spatial approach (Figure 2). Specifically, 8 out of the 11 clades exhibited a positive association (Supplementary Fig. 15). We have included this new analysis in the methods (Page 12, Line 420 to 426) and provided the results in both the main (Page 5, Line 148 to 156) and supplementary text (Supplementary Extended sensitivity analyses). We decided against running an OU climate model because we deemed it outside the scope of our study. Our main focus is on identifying the primary drivers of speciation rates across geography, and we do not attempt to explore how paleoclimate influences trait evolution given the limitations

discussed above. Nonetheless, we acknowledge that investigating this aspect could provide valuable insights for future studies.

(4) Reviewer 1: *The authors have cited the paper <https://onlinelibrary.wiley.com/doi/full/10.1111/evo.14517>, but have not utilized MiSSE, the method proposed in that paper. As the results of diversification rate estimation can vary significantly depending on the method used (this is acknowledged by the authors), it is crucial to compare more than just two metrics. MiSSE could serve as a valuable third option as it functions as an intermediate between BAMM and ClaDS, thereby becoming a more flexible model-based approach that can identify areas of rate heterogeneity or homogeneity at the tips of the tree. For these same reasons, I'm not comfortable with this premise: "we focus further analyses on BAMM speciation rates." I think results from all analyses need to be reported in tandem.*

Reply: We have addressed your suggestion by incorporating the MiSSE method as a third metric alongside BAMM and DR (Page 9, Line 290 to 294). The resulting spatial patterns of speciation were highly congruent with those observed using BAMM speciation rates (Supplementary Fig. 4). Additionally, the MiSSE method recovered the relative importance of the same predictors, namely rates of body size evolution (Supplementary Fig. 5).

Given that the BAMM metric has been previously demonstrated to outperform other metrics^{8,9} and enables direct comparison to rates of trait evolution, which are also estimated from BAMM, we chose to focus on this metric. Notably, our results using BAMM were qualitatively similar to those

obtained with DR and MiSSE (Page 10, Line 323 to 327). While we acknowledge the importance of presenting results from all three metrics together, doing so would significantly increase the number of panels and figures, making the paper overly dense. However, to provide a comprehensive overview, we have included the results from DR and MiSSE in the supplementary material (Supplementary Sensitivity Analyses; Supplementary Figs. 4 and 5), ensuring they are still accessible to interested readers.

(5) Reviewer 1: *Finally, I would like to applaud the authors for dedicating a section of the Results to account for uncertainty and sensitivity (section 3.3). While this is a great initiative, I believe this section could benefit from a more thorough interpretation of the evolutionary results in light of this uncertainty. This is particularly relevant since some trees produce highly divergent BAMM results (Figure S2): e.g., “the ray-finned fish supertree of 57, rho ranged from 0.61 to 0.64; Supplementary Fig. 2b-c.*

Reply: Thank you for acknowledging the efforts put into the Sensitivity Analysis section of our manuscript. To ensure the robustness of our findings, we conducted comprehensive sensitivity analyses to address various technical and methodological artifacts (Page 5, Line 148 to 156). These analyses specifically accounted for uncertainties in estimating rates of speciation and morphological evolution (Supplementary Figs. 4-13), the influence of paleoclimate (Supplementary Figs. 14-15), and the potential impact of biological outliers (Supplementary Figs. 16-19). In order to adhere to the recommended manuscript length standards, we have dedicated an entire section in the

supplementary material to thoroughly discuss each of the sensitivity steps undertaken and provide comprehensive interpretations of the outcomes (see Supplementary Sensitivity Analyses section), including a discussion on variance in estimates produced for alternative trees (*'Speciation rate estimates, alternative methods, and phylogenetic uncertainty'* subsection).

(6) Reviewer 1: *Regarding the morphological analyses, the use of a single approach (BAMM 'trait') may also be overly simplistic. Comparing the BAMM results with BayesTraits v4, which incorporates rate heterogeneous models, would provide a more comprehensive analysis.*

Reply: Thank you for your valuable suggestion! We have taken it into consideration and conducted additional analyses using BayesTraits in conjunction with BAMM 'trait'. We compared the tip rate evolution estimates obtained from both BAMM and BayesTraits, and found a high correlation ranging from Pearson's $r = 0.87$ to 0.90 among the five traits (Supplementary Figs. 8-12). This result indicates that our estimates of rates of morphological evolution were not biased toward a specific metric. We have now included a detailed description of BayesTraits in both the Methods (Page 11, Line 369 to 374) and Results (Page 5, Line 148 to 156) sections, as well as a comparison with BAMM, in the supplementary material to provide a comprehensive overview of our analyses (Supplementary Extended sensitivity analyses; Supplementary Figs. 8-12).

(7) Reviewer 1: *In section 2.5, the authors tested for associations among recent speciation rates and biotic and abiotic factors using multiple linear regression. However, it does not appear that they used phylogenetic regression analyses. It's important to note that even when phylogeny-based tip rates are used, an OLS regression analysis does not account for phylogenetic non-independence. This is a common mistake. I'm not sure how to perform a phylogenetically-corrected multi-linear regression analysis, but I believe there are options available. For linear regression analyses, I use `phylolm` and start by comparing OLS (no phylogeny or star phylogeny) vs. PGLS-BM (Brownian motion) vs. PGLS-OU (OU) models to find the best-fit model. Then, I report only regression analyses based on the best-fit models (sometimes a poor-fit model may produce a better *p*-value, but that's just an artifact due to model misspecification). It's crucial to follow all these steps for each regression analysis conducted, as it's not safe to assume that the best-fit model for one regression is also the best-fit model for another regression. Find a way to apply the same rationale to phylogenetically-corrected multi-linear regression analyses.*

Reply: We appreciate the suggested method and the clear explanation provided. It was helpful to learn how to apply this analysis. However, we have concerns about the suitability of using a phylogenetic approach such as PGLS for our spatially-focused analyses of sub-basins. The typical application of PGLS is at the species level, which may not align with our specific approach focused on the sub-basin level. We apologize if we did not make this clearer in the main text and have made edits to the Introduction accordingly (Page 3, Line 75 to 90).

(8) Reviewer 1: 2.3 | *Rates of morphological evolution and species diversity.*”

This subsection covers three different topics: the collection of morphological traits, the estimation of rates of morphological evolution, and the species diversity per basin/species area relationship. Consider splitting it into multiple subsections to improve the flow. In fact, the last paragraph on species diversity seems more connected with the previous section, as it is related more to speciation rates than to morphology: “overall, we found that species density and area predicted speciation rates similarly.

Reply: Thank you for this observation. We split the former “section 2.3” into two new subsections. One describes traits and their rates of evolution (Page 10, Line 339 to 382), while the other describes species diversity (Page 11, Line 384 to 403).

(9) Reviewer 1: 2.4 | *Abiotic variables.*” *Same thing here. The title of the subsection leaves the impression that it is only about collection of abiotic variables. But there’s also analyses linking abiotic variables with speciation rates. Perhaps just add one subsection linking biotic (morphology) and abiotic factors with speciation rates.*

Reply: We agree that the initially proposed section included some additional procedures not entirely related to extracting abiotic variables. However, these procedures are important for the validation of the selected variables. Therefore, we have split the “Abiotic variables” section into two separate sections: “Climate variables” (Page 12, Line 405 to 426) and “Habitat variables” (Page 13, Line

428 to 441). We believe this new organization is more appropriate and addresses the reviewer's concern.

(10) Reviewer 1: *I feel that the M&Ms and Results sections in the main text are too condensed. The supplement would benefit from an expanded explanation of the research approach and findings in these sections.*

Reply: We agree with your suggestion that more detailed information regarding the Methods should be provided. We have expanded the Methods section in both the main text (highlighted in blue) and supplementary material, providing more detailed information about the methods employed. Furthermore, we have inserted in the supplementary material a section dedicated to Sensitivity Analyses, in which we provide more detailed information and a more in-depth discussion of the approaches taken. This additional information should make it easier for readers to understand the technical aspects of our research (see Supplementary Extended methods and sensitivity analyses).

(11) Reviewer 1: *Typo: "in which are more accurate than deep-time estimates."*

Reply: We have made revisions to the sentence and improved its phrasing (Page 9, Line 290 to 294).

(12) Reviewer 1: *Figs. S16–S18. Do discuss something about the outliers in these plots. Also, why report r rather than r square values? And p values?*

Reply: Additional panels have been included to display the speciation-temperature relationship after the removal of the five prominent outliers (Supplementary Fig. 14). Overall, these associations remained qualitatively similar. It is worth noting that our conclusions did not change when we conducted a multiple linear regression, which also excluded these outliers, and yielded consistent results (Supplementary Figs. 18-19). Furthermore, we have provided r and p values for all necessary captions in the supplementary material. To showcase the strength of the association trend between the variables involved, we have opted to use Pearson's correlation coefficient instead of R-squared.

(13) Reviewer 1: *The maps displaying rates of evolution are some of the most striking results of this paper. It would be beneficial to include a map with rates of body size evolution in the main text. This could be achieved by adding a new figure or panel to Figure 1.*

Reply: We agree that the addition of a new panel displaying the spatial patterns of body size evolution would significantly enhance the comprehensibility of our results. Accordingly, we have inserted this new panel into Figure 1 (Figure 1e).

(14) Reviewer 1: *The figures in the PDF I downloaded in the main text (but not the SM file) have extremely low resolution (cannot read axis labels in many plots). Make sure high-resolution figures are uploaded for the final version.*

Reply: I think there was a technical inconsistency when the uploaded document was converted to PDF by the system. The figures in the original file have high resolution. However, we have confirmed that the figures uploaded in the supplementary material are now of good resolution. We apologize for this mistake.

(15) Reviewer 1: *The trees plotted on the SM file need taxonomic annotations.*

Reply: Thank you for noticing. We added taxonomic groups to the trees in the supplementary material.

(16) Reviewer 1: *On the contrary, evolutionary rates of oral gape position and relative maxillary length were negatively related to speciation rates (Fig. 2 and Figs. 3b-c; Table 2). Body elongation evolution had no significant association with speciation rates (Fig. 2a; Table 2). We also observed a weak, but significant, positive relationship between speciation rates and species diversity (Fig. 2 and Fig. 3d; Table 2).” Add R2 values within the parentheses.*

Reply: We have included the R-squared values inside the parentheses as suggested (Page 4, Line 121 to 130). Thank you!

(17) Reviewer 1: *Label all supplementary figures as “Fig. SX” or “Supplementary Fig. X.” Same for tables.*

Reply: We modified the supplementary figures as you suggested by relabeling them from “Fig. X” to “Figure SX”.

Comments from reviewer 2 and respective replies or actions taken:

(1) Reviewer 2: *The authors have aggregated an impressive dataset from relevant databases and used a newly published tree to investigate ecological, geographic, and environmental drivers of speciation rates in diverse vertebrate radiation. The taxonomic scale of the analyses is impressive, and the authors include a robust set of sensitivity analyses to support their methods and results. The finding that high elevation and rapid body size evolution drive high speciation is novel and of broad interest to evolutionary biologists. However, I have identified some weaknesses and flaws that make it hard to determine whether their findings are accurate. Therefore I recommend major revisions to this manuscript.*

Reply: Thank you for your feedback, and we are delighted to hear that you found our study to be novel and enjoyable. We appreciate your valuable input, and we agree that further revisions could enhance the clarity and quality of the manuscript. Accordingly, we have taken your comments into consideration and worked to make the necessary revisions to the manuscript.

(2) Reviewer 2: *While BAMM is appropriate for their analyses, the estimates of tip speciation rates across ~2300 species raise some concerns. First, the rate shifts identified by BAMM are very clade-specific and do not appear to recover known rate shifts within clades from other published studies. These include Cichliformes; Geophagini, and Heroini (see Burress & Tan 2017), and Characidae and Anostomidae (see Melo et al. 2022). The authors should*

compare the estimates of speciation rates in this study with the above-referenced studies as they did for the Poecillidae tree (line 137). At the very least, they need to address why well-known lowland lineages that show elevated speciation rates in other studies using BAMM do not show high rates in their analysis.

*Burress, Edward D., and Milton Tan. "Ecological opportunity alters the timing and shape of adaptive radiation." *Evolution* 71.11 (2017): 2650-2660.*

*Melo, Bruno F., et al. "Accelerated diversification explains the exceptional species richness of tropical characoid fishes." *Systematic Biology* 71.1 (2022): 78-92."*

Reply: We conducted BAMM analysis for the suggested phylogenies in addition to the clade-specific tree of Poecilia. We found a moderate correlation between the BAMM tip rates recovered from our study and the species estimates of Burress & Tan, 2017 tree¹⁰ ($r = 0.633$, $p < 0.001$) and Melo et al., 2022 tree¹¹ ($r = 0.655$, $p < 0.001$). We have included these results in the supplementary material (Supplementary Fig. 7) and have also discussed potential explanations for the variance in estimates that deviated from our observed estimates (see Supplementary Sensitivity Analyses).

(3) Reviewer 2: *Secondly, BAMM speciation rates appear to vary slightly across most of the phylogeny, and exceptionally high rates are concentrated in a few small clades. I am concerned that the exceptional rate of these small clades, particularly Orestias, may impede the ability to detect rate shifts of a smaller magnitude within other clades. Therefore, I recommend removing all*

Orestias species and repeating the BAMM analyses. Also, it would be helpful for the authors to report the distribution of tip rates in the supplementary materials.

Reply: Thank you for the suggestion. We agree that speciation rates in *Orestias*, a small clade, may affect estimations in other clades. Thus, we removed *Orestias* species from the phylogeny and reran BAMM analysis (Page 14, Line 470 to 479). BAMM tip speciation estimates were congruent with and without *Orestias* species (Page 5, Line 148 to 156), indicating that the exceptionally high rates observed in *Orestias* did not hinder the detection of rates in other clades (Supplementary Fig. 16). In addition, we reported the density of tip rates in the supplementary material at both the species and sub-basin levels (Supplementary Figs. S16 and S18).

(4) Reviewer 2: *Similarly, in Supplementary Figure 2D, the speciation rates calculated from the Reznick et al. 2017 tree vary on a scale much greater than those calculated in this study, again suggesting that the BAMM analyses may be underestimating rate variation within clades.*

Reply: While we acknowledge that the tree presented by Reznick et al., 2017¹² exhibits greater variation compared to our study, it is important to note that their phylogeny includes a smaller number of species (n = 294 species). This could potentially result in an overestimation of speciation rates. Nonetheless, our study has produced consistent speciation rate estimates across different scenarios, including using a more representative tree from Casemiro et al., 2023, and removing smaller clades with the highest speciation rates, among

others. However, we have addressed these concerns as caveats in our study, which can be found in Supplementary Sensitivity Analyses (*'Speciation rate estimates, alternative methods, and phylogenetic uncertainty'* subsection).

(5) Reviewer 2: *The authors take the mean speciation rate of all co-occurring species within each sub-basin, which I believe is problematic because of the extreme variation in rate between lineages. They should also report the range or distribution of rates within each sub-basin. Because rates appear to vary little across most of the tree (excluding a few outlier clades), including even one or two exceptional species may be enough to inflate the mean rate of a sub-basin with low species richness. Reporting the min and the max rates would at least show whether the overall distribution of rates varied across basins. Also, I suggest performing the multiple regression analysis across the min and max values.*

Reply: We reported the distribution of rates in each sub-basin by calculating the median, minimum, and maximum statistics. Our findings revealed that speciation rates vary across South America, as indicated by median, minimum, and maximum speciation rates. Median speciation rates exhibited spatial patterns similar to those using mean speciation rates. Furthermore, mapping the range of speciation rates, such as using minimum and maximum values, allowed us to observe that sub-basins in the Amazon basin had the lowest speciation rates, while higher rates were concentrated at high elevations (Supplementary Fig. S20).

Additionally, we conducted multiple linear regressions using median, minimum, and maximum speciation rates as response variables instead of mean values. These analyses supported our conclusions, particularly when utilizing median estimates (Supplementary Fig. S21). Body size evolution emerged as the strongest predictor, followed by elevation. All these results were inserted in both the main text (Page 9, Line 311 to 322) and supplementary material (Supplementary Figs. S20-S21).

(6) Reviewer 2: *Similarly, the authors did not account for the geographic range of species. Wide-ranging species with exceptional speciation (either high or low) rates may bias estimates. The authors did remove sub-basins with <5, <10, <15, and <20 species; however, they do not report on the significance of these results and make only qualitative comparisons (line 329). Similarly, the authors remove the 5 outlier sub-basins containing exclusively Orestias species; however, it is unclear whether Orestias species occur in other sub-basins and if the authors removed those. Because Orestias rates are exceptionally high, they may still bias the mean speciation rate in any sub-basin they are in, even if that basin is relatively speciose. If the authors did remove all sub-basins with Orestias species before repeating the multiple linear regression (Supplementary Table 3), they should clarify that. In any case, I suggest identifying all sub-basins in which Orestias species occur and removing Orestias from all the sub-basins.*

Reply: We acknowledge the concern that wide-ranging species may excessively influence our estimates. To address this, we calculated the average

evolutionary rates for each sub-basin by weighting them based on the inverse of species range size, as done in previous macroecological studies^{1,13}. For example, species that exhibit exceptionally high speciation rates and have wide distributions will be given less weight in our averaged estimates. To determine species range size, we summed the area of all sub-basins where the species occurs. We then divided the evolutionary rates of each species, including speciation and morphological evolution rates, by their respective range sizes. When accounting for species range size, our findings remained consistent with our main results (Supplementary Fig. 36), indicating that wide-ranging species did not bias our sub-basin level estimates. These results are presented in both the main text (Page 12, Line 397 to 403) and supplementary material (Supplementary Fig. 36).

Additionally, we excluded sub-basins with less than 10, 15, and 20 species to investigate whether the observed patterns were influenced by regions with low species richness. Our findings remained consistent after this exclusion (Supplementary Figure 35), and we now present a quantitative analysis of these results (Page 12, Line 397 to 403).

Finally, we addressed the issue of *Orestias* species in three different ways (Page 14, Line 470 to 479). Firstly, we excluded *Orestias* species from the phylogeny and re-ran BAMM analysis, resulting in highly correlated tip rate estimates compared to when *Orestias* species were included (Supplementary Figure 16). Secondly, we identified all sub-basins occupied by *Orestias* species and excluded these species from the calculation of average speciation rates. This approach also did not alter our conclusions (Supplementary Fig. 17). Thirdly, we removed the five outlier sub-basins that were predominantly

occupied by *Orestias* species, but found no significant impact of their removal on our main findings (Supplementary Figs. 18-19). Taken together, we are confident that our findings are robust against species with exceptional speciation rates, and such rates do not compromise our estimates at the phylogenetic level or when averaging estimates across space (Page 5, Line 148 to 156).

(7) Reviewer 2: *Lastly, the color scale in Figure 1 B and C is misleading. The range of the highest category (red) is greater than the range of all other categories. The lower end of the highest category (~0.18) may not significantly differ from most other basins (0.12-0.18). In contrast, the high range of 0.61 is significantly higher than the lower range of the same category. This color scheme makes it look like all red basins have a high speciation rate when they might only be .01 different from another sub-basin. The authors should redo the figure with more equal ranges between colors.*

Reply: Thank you for noticing the issue with the color scale (panels 1b and 1c). As you suggested, we reformulated the figure with more intervals to improve clarity in visualizing speciation differences across sub-basins (see Fig. 1).

References mentioned in the response letter

1. Jetz, W., Thomas, G. H., Joy, J. B., Hartmann, K. & Mooers, A. O. The global diversity of birds in space and time. *Nature* **491**, 444–448 (2012).
2. Vasconcelos, T., O'Meara, B. C. & Beaulieu, J. M. A flexible method for estimating tip diversification rates across a range of speciation and extinction scenarios. *Evolution* **76**, 1420–1433 (2022).
3. Venditti, C., Meade, A. & Pagel, M. Multiple routes to mammalian diversity. *Nature* **479**, 393–396 (2011).
4. Cassemiro, F. A. S. *et al.* Landscape dynamics and diversification of the megadiverse South American freshwater fish fauna. *Proc. Natl. Acad. Sci.* **120**, e2211974120 (2023).
5. Karger, D. N. *et al.* Climatologies at high resolution for the earth's land surface areas. *Sci. Data* **4**, 170122 (2017).
6. Morlon, H. *et al.* RPANDA: an R package for macroevolutionary analyses on phylogenetic trees. *Methods Ecol. Evol.* **7**, 589–597 (2016).
7. Zachos, J. C., Dickens, G. R. & Zeebe, R. E. An early Cenozoic perspective on greenhouse warming and carbon-cycle dynamics. *Nature* **451**, 279–283 (2008).
8. Title, P. O. & Rabosky, D. L. Tip rates, phylogenies and diversification: What are we estimating, and how good are the estimates? *Methods Ecol. Evol.* **10**, 821–834 (2019).
9. Cooney, C. R. & Thomas, G. H. Heterogeneous relationships between rates of speciation and body size evolution across vertebrate clades. *Nat. Ecol. Evol.* (2020) doi:10.1038/s41559-020-01321-y.
10. Burress, E. D. & Tan, M. Ecological opportunity alters the timing and shape of adaptive radiation. *Evolution* **71**, 2650–2660 (2017).
11. Melo, B. F. *et al.* Accelerated Diversification Explains the Exceptional Species Richness of Tropical Characoid Fishes. *Syst. Biol.* **71**, 78–92 (2022).

12. Reznick, D. N., Furness, A. I., Meredith, R. W. & Springer, M. S. The origin and biogeographic diversification of fishes in the family Poeciliidae. *PLOS ONE* **12**, e0172546 (2017).
13. Harvey, M. G. *et al.* The evolution of a tropical biodiversity hotspot. *Science* **370**, 1343–1348 (2020).

REVIEWERS' COMMENTS

Reviewer #1 (Remarks to the Author):

I commend the authors for carefully addressing all the points I raised earlier. The manuscript is no doubt much improved after the revision. Looking forward to seeing this published!

Reviewer #2 (Remarks to the Author):

I commend the authors on their thorough job addressing my comments. I have no additional comments and believe the manuscript is now suitable for publication.